# Assessing the Repeatability and Reliability of NIRS to Predict Nutritional Values and to Evaluate Two Lignin Methods in *Urochloa* spp. Grasses

Iuli Caetano da Silva Brandão Guimarães [1] , Thiago Henrique da Silva [2,*], Cristina Cirino Picchi [3] and Romualdo Shigueo Fukushima [1]

1 Department of Animal Nutrition and Production, School of Veterinary Medicine and Animal Science, University of Sao Paulo, Pirassununga 13635-900, SP, Brazil
2 Department of Animal Science, School of Animal Science and Food Engineering, University of Sao Paulo, Pirassununga 13635-900, SP, Brazil
3 Embrapa Pecuária Sudeste, São Carlos 13560-970, SP, Brazil
* Correspondence: silvath@usp.br; Tel.: +55-(19)-3565-4200

**Abstract:** Reliable forage analysis is crucial for proper ration formulation of ruminant herds. Through its fast, inexpensive, and non-destructive procedures, near-infrared spectroscopy (NIRS) has become a valuable method for forage evaluating. Notwithstanding, NIRS needs calibration before routine analysis. In addition, to evaluate the best method for lignin quantification in *Urochloa* spp. grasses is crucial under a digestibility perspective in grass-fed ruminant production. The aims of this study were to use 149 samples from different *Urochloa* species to develop NIRS calibration curves (partial least squares regressions) for acid detergent lignin (ADL), acetyl bromide lignin (ABL), as well as for ash, cell wall (CW), neutral detergent fiber (NDF), acid detergent fiber (ADF), in vitro DM digestibility (IVDMD), and in vitro NDF digestibility (IVNDFD). Moreover, the aim of this study was to correlate the in vitro digestibility with lignin quantification methods: ADL and ABL. Near-infrared spectroscopy showed potential for the quantification of *Urochloa* spp. properties, such as lignin contents (ADL and ABL) and ash, CW, NDF, ADF, IVDMD, and IVNDFD. However, calibrations performed using NIRS to measure ADF, ADL, IVDMD, and IVNDFD need to be thought about with caution before their utilization as a routine analysis for determining the potential for nutrient measurement and digestibility of *Urochloa* spp. grasses. In addition, the ABL method used for lignin quantification was better correlated with IVDMD and IVNDFD than the ADL method.

**Keywords:** *Brachiaria* spp.; chemometry; forage quality; partial least squares regressions; tropical grass

## 1. Introduction

In some countries, beef and dairy herds are traditionally based on grazing of *Urochloa* spp. grasses. These grasses can adapt to drought and low-fertility soils as well as possess a high carbon-sequestration capacity and improved efficiency of nitrogen utilization compared with others genus [1]. As fresh pasture, the superior nutritional quality of *Urochloa* spp. grasses is the primary goal of farmers to reach high profitability in their business [2,3]. However, the forage quality varies through biotic and abiotic factors including nutrient availability, stage of maturity, topography, and climatic conditions [4–6]. Accordingly, reliable forage analysis is crucial for proper ration formulation according to herd requirements.

Traditionally, standard reference methods (SRM) so-called wet chemistry techniques have been performed for determining nutritional values in forage. However, they are time-consuming, have a high cost, are laborious, and have a negative environmental impact; making these analytical techniques unfeasible [7–9]. Regarding forage quality, lignin content is one of the most important elements used to assess forage quality. This

is because lignin determines fiber digestibility and, consequently, forage intake by rumi­nants [10,11]. Nevertheless, inconsistencies have been reported among the most important methods to assess lignin content in forages [12,13]. For example, partial solubilization of lignin has been detected through the acid-detergent lignin (ADL) procedure [14,15], which may underestimate the lignin content in forage compared to acetyl-bromide lignin (ABL) [13]. Inputting erroneous lignin values into the cattle feed software formulation may impair fiber digestibility prediction, and as consequence, negatively impact the accuracy of ruminant nutrition.

The near-infrared spectroscopy (NIRS) technique has been applied since the 1960s in many different areas of science, such as neurology, feed, and raw material from pharma industries [16]. Moreover, after the Norris et al. [17] study, this technique has become widely recognized as a valuable method for forage evaluation. Separately from SRM, the NIRS prediction analysis, does not require sample processing or reagents, which makes it a faster and more cost-effective procedure [18]. Thus, according to the positive factors pointed out, the NIRS procedure represents a radical shift from conventional chemical analysis, being a rapid decision tool which also allows large-scale analysis.

Few studies have reported NIRS calibration models for predicting nutritional values in *Urochloa* spp. grass populations [16,19,20]. Therefore, this study aimed to assess the repeatability and reliability of NIRS to predict nutritional values in *Urochloa* spp. grasses. In addition, we have evaluated the lignin-NIRS prediction in *Urochloa* spp. by carrying out ABL and ADL methods. We hypothesized that NIRS would be able to predict nutritional values of *Urochloa* spp. accurately. Further, the lignin-NIRS prediction would be improved by performing the ABL instead of the ADL method.

## 2. Materials and Methods

Standard reference methods (SRM) were carried out at the Laboratory of Feed Analysis of the Department of Animal Nutrition and Production of the School of Veterinary Medicine and Animal Science of University of São Paulo, Brazil. Near-infrared spectroscopy analyses were undertaken at the Laboratory of Animal Nutrition of Embrapa Pecuária Sudeste, Brazil. All the experimental procedures were approved by the Bioethics Committee of the School of Veterinary Medicine and Animal Science, University of São Paulo (approval number: 7684280518).

### 2.1. Urochloa spp. Samples

A total of 169 *Urochloa* spp. samples of different cultivars were taken from the Embrapa laboratory of animal nutrition database. The samples were harvested from four cities in three states from Brazil [Campo Grande, Mato Grosso do Sul state (Midwest; $n = 61$); São Carlos, São Paulo state (southeast; $n = 58$); Juiz de Fora ($n = 26$) and Sete Lagoas ($n = 24$) both from Minas Gerais State (southeast)]. According to Köppen and Geiger, the climates from Campo Grande, São Carlos, Juiz de Fora, and Sete Lagoas may be classified as Aw, Cfa, Cwa, and Aw, respectively. In addition, the samples grown were passed by different agronomic and physiological management, such as fertilization level, maturity stage, and production systems. All these different sample locations and managements were important to improve the robustness of the NIRS calibration model since it was possible to include the extensive diversity of *Urochloa* spp. throughout an important Brazilian territory for ruminant production.

### 2.2. Laboratory Analysis

2.2.1. Standard Reference Methods (SRM)

Grass samples from four different regions in Brazil were harvested and dried at 60 °C using a forced-air oven for 72 h. Then, the samples ($n = 169$) were ground to pass a 1-mm screen (Willey mill; MARCONI®-MOD-0.48) and stored in sealed containers for further chemical analyses and digestibility assays. Dried and ground grass samples were analyzed in duplicates for DM at 100 °C (method 967.03; [21]), ash (method 942.05; [21], NDF (TE-149

fiber analyzer, Tecnal Equipment for Laboratory Inc., Piracicaba, Brazil; [22], and ADF (method 973.18; [23]. For CW analyses, the ground samples were sequentially submitted to water, ethanol, chloroform:methanol (2:1), and acetone by using Soxhlet equipment. For each of the four solvent phases, the extraction time lasted about 4 to 8 h [24].

Lignin contents were performed by using two methods: acid-detergent lignin (ADL) and acetyl-bromide lignin (ABL). The ADL method was performed according to 25, in which the samples were soaked in 12 *M* sulfuric acid for 3 h and thoroughly washed with boiling distilled water. Then, the bags were dried at 105 °C and weighed. Later, for lignin ash-free, the bags were burned using a muffle at 600 °C [25].

The ABL determination followed the procedure outlined by Fukushima [13]. This method consists of digesting 100 mg of the CW preparation with 4.0 mL of a 25% acetyl bromide in acetic acid reagent at 50 °C for 2 h, with occasional mixing. After cooling, the volume was made up to 16.0 mL with acetic acid (HAc) and centrifuged ($3000 \times g$, 15 min). A half milliliter of this solution was added to a tube containing 2.5 mL of HAc and 1.5 mL of 0.3 M NaOH. After shaking, 0.5 mL of 0.5 M hydroxylamine hydrochloride solution was added, and the volume made up to 10 mL with HAc. Optical density at 280 nm was measured and concentration determined from the respective standard curve using a regression equation. A blank was included to correct for reagent background absorbance. The regression Equation (1) was set up encompassing the mean of molar absorptivity coefficients from 17 lignin purified samples extracted with acidic dioxane, as follows:

$$X = (Y - 0.0009)/23.077, \tag{1}$$

where, X is the lignin content (mg/mL); Y is the optic density from an unknown sample; 0.0009 is the intercept mean; and 23.077 is the molar absorptivity coefficient from 17 lignin calibration curves [26]. The X value is multiplied by CW content and divided by the initial sample weight.

### 2.2.2. In Vitro Digestibility

In vitro dry matter digestibility and IVNDFD were determined according to Tilley and Terry [27]. Briefly, duplicate samples (*n* = 169) from oven-dried forage (0.5 g) were weighed in Ankon bags, which were previously dried and weighed. In the test tubes, 40 mL of McDougall solution (artificial saliva) was added to 10 mL of rumen inoculum of steers kept grazing *Urochloa decumbens* pasture, supplemented with 3.0 kg of corn silage (DM basis) and ad libitum mineral salt. Tubes were sealed with rubber corks containing a Bunsen gas release valve, immediately after flushing out with $CO_2$ and incubated in oven for 48 h under controlled temperature (39 °C). They were agitated at least three to four times during fermentation. The second IVDMD phase occurred after discarding the liquid solution. Pepsin solution (1:10,000) at 0.2% (50 mL) was added to each tube, followed by agitation at 39 °C for another 48 h. After washing, drying, and weighing the bags, calculations were performed as the Equation (2) below:

$$\text{IVDMD} = [100 \times \text{g of DM in sample} - (\text{g of residual DM} - \text{g of DM of inoculum without sample})]/\text{g of DM in sample} \tag{2}$$

The same procedure was performed in IVNDFD analysis, except that the bags were submitted to NDF washing procedures after incubation.

### 2.3. NIRS Procedures

The grass samples (*n* = 169; *n* = 149 for model production and *n* = 20 for prediction) dried at 60 °C and milled through a 1-mm sieve were used for NIRS scanning. All samples were redried at 100 °C for 4 h to remove effects of residual moisture and allowed to cool in a desiccator before being scanned in a borosilicate glass Petri dish PyrexTM on a BÜCHI NIRFlex *n*-500 solids spectrophotometer (BÜCHI Labortechnik, Flawil, Switzerland) in a temperature-controlled room (20 °C ± 1). Samples were scanned in duplicate from 10,000

to 4000 cm$^{-1}$ with a resolution of 4 cm$^{-1}$ and 32 scans per spectrum (Figure 1). The spectral reflectance curves were obtained using diffuse reflectance due to the Operator 1.2 software (BÜCHI Labortechnik, Flawil, Switzerland).

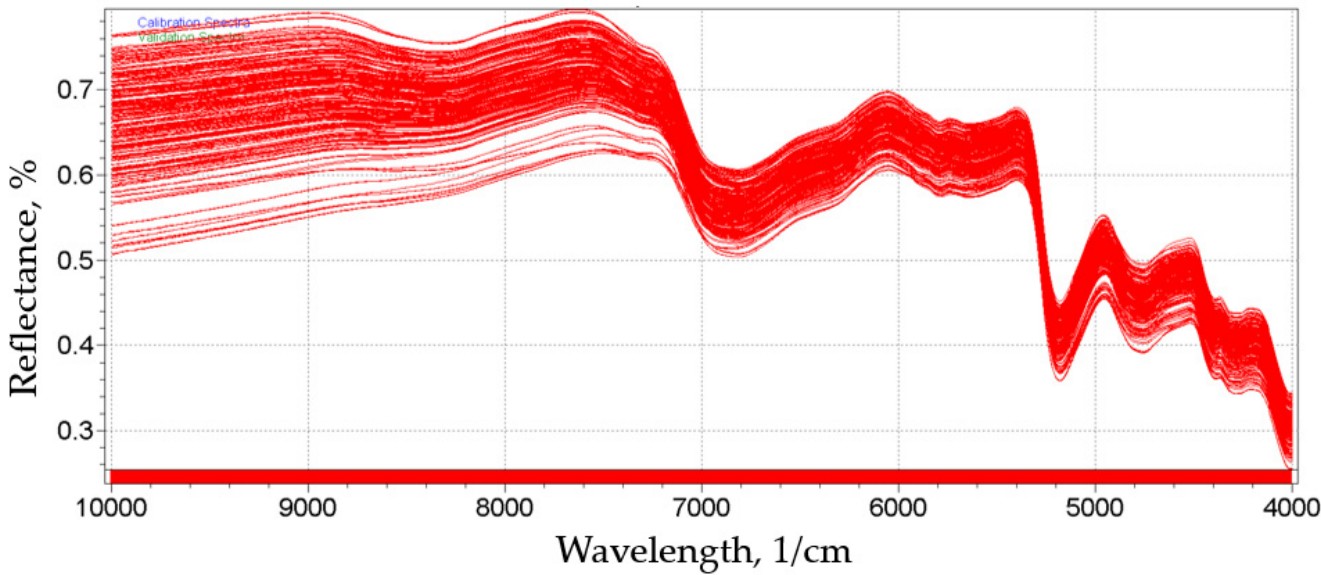

**Figure 1.** NIRS spectra obtained from samples of *Urochloa* spp. (*n* = 149) grown under different edaphoclimatic conditions.

*2.4. Statistical Analysis*

Multivariate Calibration

All calibrations (*n* = 149) were performed from a wavelength spectral range of 8000 to 4100 cm$^{-1}$ using diffuse reflectance. The spectrophotometer equipment was automatically calibrated using the SpectralonTM pattern every 10 min. In order to mitigate the light scattering due to the heterogeneity of sample size particles and noises, as well as to increase the small peaks of absorption in the spectrum (sensibility) a selection of wavelengths was performed, and several pre-treatments were used before the calibration. Different pre-treatments were performed such as normalization, smoothing, and derivatives. For normalization, the data pre-treatments n01 (normalization between 0 and 1), mf (Multiplicative Scatter Correction [full]), and SNV (Standard Normal variate) were carried out. For smoothing, only the sa3 (smoothing 3 points) data pre-treatment was performed, and, finally, for derivatives, the data pre-treatments dg1 and dg2 (first and second derivatives Savitzky-Golay 9 points) were executed.

The best calibrations were compared using the Multiplicative scatter correction full (mf) pre-treatment, except in the case of CW, where mf was the pretreatment of choice. It was then compared to the Smoothing Average 3 Points (sa3) pre-treatment, which was the second best and as satisfactory results as mf.

Further, after outlier exclusion, the models were evaluated as their content range, coefficient of determination (R$^2$), and standard error of calibration (SEC) and validation (SEP). All chemical and nutritional values assessed using SRM were included in the NIR equipment to set up the multivariate calibration curve using the NIRCal 5.6 software (BÜCHI Labortechnik, Flawil, Switzerland).

All spectra (X matrix) were linearly combined with the reference values (Y matrix) through the Partial Least Squares Regression (PLS) method. Afterward, the Principal Components Analysis (PCA), set up vectors (factors) that are uncorrelated to each other and are able to explain the variation of the data of each matrix. Finally, the matrices are related in order to maximize the covariance between the two datasets [28]. This method aims to find a relationship between the matrix (X) containing the spectra of the samples

and the vector that stores the respective concentrations (y). The result is an equation similar to Equation (3):

$$y = Xb + e, \qquad (3)$$

where, b is the regression vector and e is the vector representing the model errors.

Therefore, the selection of samples for prediction (external validation) was performed through the Euclidean distance to select the representative samples of variability by using the Kennard-Stone algorithm [29]. One hundred and forty-nine samples (88.2%) were used for model production and 20 samples (11.8%) for prediction. Still, regarding prediction, the minimum and maximum values were within the model range (calibration range).

Data were analyzed using SAS 9.4 (SAS Institute Inc., Cary, NC, USA). Before the actual analysis, data were explored to seek disparate information ("outliers") and for normality of residuals using the Shapiro-Wilk test (PROC UNIVARIATE). An individual observation was considered an outlier when standard deviations about the mean or to the model were bigger than +3 or lesser than −3. Descriptive statistics analysis was undertaken using the PROC MEANS procedure. For simple regression models and Pearson's correlation analyses, the PROC CORR and PROC REG procedures were, respectively, used.

The spectral validation from NIRS was assessed using the Partial Least Squares Regression (PLS). The outlier detection and their extraction from calibration curves were performed using the "outlier detection" tool in the NIRCal 5.6 software (BÜCHI). The model performance was assessed according to several indexes, such as (1) Q value; indicating the calibration quality. Values greater than 0.75 are acceptable for quantitative analyses. However, values between 0.5–0.75 are inaccurate, and values lower than 0.5 should be evaluated before their applications. This index may be used to choose the best calibration model; (2) SEC; (3) SEP, which must be as low as possible and closest to SEC value; (4) correlation (r); (5) coefficient of determination ($R^2$); (6) Bias (systematic error; which must not be greater than 0.6 times the SEC; and finally; (7) standard error. High Q, r, and $R^2$ and low SEC, SEP, and root mean of standard error of prediction (RMSEP), indicate the best prediction model. The RMSEP value exhibits the concordance between the estimate and the reference values.

For external validation assessment (prediction), the NIRCal 5.6 (BÜCHI®) software included new statistical parameters, which were: offset, slope, RMSEP, SEP, residual standard deviation (RSD), and bias. In this study, the external validation was performed using an independent test set without outlier values (*n* = 20) to ensure relevant estimates.

Root mean of standard error of prediction, SEP, and RSD are different estimates of random error, which include: RMSEP = random error + systematic slope + bias; SEP = random error + systematic slope; and RSD = random error. Hence, RMSEP is a more complete estimate.

## 3. Results

### 3.1. Descriptive Analyses

The chemical and nutritional contents of *Urochloa* spp. of the 169 samples obtained using standard reference methods are presented in Table 1. High variability was detected for ADF, ADL, ABL, IVDMD, and IVNDFD, which demonstrated a coefficient of variation of 12.0, 22.1, 17.9, 15.5, and 16.7, respectively. The variation of all chemical and nutritional items herein evaluated could be considered acceptable and broad enough for the development of adequate NIRS calibration models.

The same dataset of chemical and nutritional values assessed using SRM presented in Table 1 (*n* = 169), was split into 2 groups for NIRS models calibration and its internal validation (*n* = 149), as well as for its external validation (*n* = 20; Table 2). In Table 3, the chemical and nutritional values assessed using NIRS of the same samples used for external validation using SRM in Table 2 is shown. These values presented in Table 2 (for external validation using SRM) and in Table 3 (for external validation using NIRS), were used for assessing the correlation and association of chemical and nutritional values between both techniques.

**Table 1.** Descriptive analyses of nutritional values of *Urochloa* spp. samples performed using standard reference methods (*n* = 169; g/kg DM).

| Analyte | Mean | Minimum | Maximum | SD [1] | CV [2] |
|---|---|---|---|---|---|
| Ash | 76.05 | 40.77 | 124.98 | 17.60 | 23.1 |
| Cell wall | 809.82 | 670.12 | 916.44 | 53.13 | 6.56 |
| Neutral detergent fiber | 761.52 | 575.42 | 879.24 | 60.74 | 7.98 |
| Acid detergent fiber | 489.14 | 314.31 | 611.55 | 58.81 | 12.0 |
| Acid detergent lignin | 71.57 | 37.88 | 111.52 | 15.83 | 22.1 |
| Acetyl bromide lignin | 114.77 | 61.12 | 173.9 | 20.54 | 17.9 |
| IVDMD [3] | 602.73 | 331.65 | 791.62 | 93.31 | 15.5 |
| IVNDFD [4] | 543.26 | 298.91 | 740.17 | 90.88 | 16.7 |

[1] Standard deviation; [2] Coefficient of variation; [3] In vitro dry matter digestibility; [4] In vitro neutral detergent fiber digestibility.

**Table 2.** Descriptive analyses of nutritional values of *Urochloa* spp. grass samples used in calibration and internal validation (*n* = 149) performed using standard reference methods (g/kg DM; otherwise stated).

| Analyte | Mean | Minimum | Maximum | SD | CV |
|---|---|---|---|---|---|
| Ash | 76.29 | 40.77 | 124.9 | 17.49 | 22.93 |
| Cell wall | 811.6 | 670.1 | 916.4 | 54.05 | 6.66 |
| Neutral detergent fiber | 768.2 | 575.4 | 879.2 | 58.84 | 7.66 |
| Acid detergent fiber | 494.2 | 314.3 | 611.5 | 57.84 | 11.70 |
| Acid detergent lignin | 102.4 | 60.93 | 151.4 | 19.86 | 19.39 |
| Acetyl bromide lignin | 113.8 | 61.12 | 173.9 | 20.32 | 17.86 |
| IVDMD [1] | 598.7 | 331.6 | 791.6 | 93.28 | 15.58 |
| IVNDFD [2] | 542.4 | 298.9 | 740.1 | 90.77 | 16.73 |

[1] In vitro dry matter digestibility. [2] In vitro neutral detergent fiber digestibility.

**Table 3.** Descriptive analyses of nutritional values of *Urochloa* spp. samples used in external validation (*n* = 20) performed using standard reference method (SRM) or near-infrared spectrophotometry (NIRS).

| Analyte | SRM | | | | | NIRS | | | | |
|---|---|---|---|---|---|---|---|---|---|---|
| | Mean | Minimum | Maximum | SD | CV | Mean | Minimum | Maximum | SD | CV |
| Ash | 74.27 | 49.84 | 107.5 | 18.7 | 25.2 | 73.3 | 47.2 | 102.5 | 15.73 | 21.5 |
| Cell wall | 795.9 | 711.9 | 859.4 | 44.35 | 5.57 | 796.1 | 701.1 | 865.4 | 47.53 | 5.97 |
| Neutral detergent fiber | 708.7 | 616.4 | 800.9 | 50.91 | 7.18 | 741.3 | 641.9 | 803.3 | 48.66 | 6.56 |
| Acid detergent fiber | 451.2 | 341.2 | 549.5 | 52.9 | 11.7 | 470.8 | 364.7 | 547.1 | 47.33 | 10.1 |
| Acid detergent lignin | 81.33 | 58.37 | 119.2 | 15.11 | 18.6 | 68.8 | 51.9 | 95.8 | 12.25 | 17.8 |
| Acetyl bromide lignin | 123.9 | 69.88 | 155.4 | 22.47 | 18.1 | 121.9 | 69.9 | 155.3 | 21.29 | 17.5 |
| IVDMD [1] | 632.3 | 445.8 | 774.1 | 90.31 | 14.3 | 639.4 | 491.6 | 784.6 | 76.37 | 11.9 |
| IVNDFD [2] | 560.1 | 372.7 | 708.5 | 99.99 | 17.9 | 567.1 | 398.8 | 774.7 | 92.37 | 16.3 |

[1] In vitro dry matter digestibility. [2] In vitro neutral detergent fiber digestibility.

### 3.2. NIRS Model Calibrations and Internal Validation

The statistical performance of calibration and internal validation models are presented in Table 4. One pretreatment was enough for CW, NDF, and ABL calibration models. For Ash, ADF, IVDMD, and IVNDFD requiring two pretreatments and, finally, ADL, which required three pre-treatments.

The number of samples in the calibration set was close to 2/3 (about 66.66%) for calibration and 1/3 (about 33.33%) for internal validation of the 149 samples selected for these procedures. As they were analyzed in duplicate, 298 spectra were analyzed (Table 4). The number of samples used in the calibration and internal validation was not similar among the analytes due to the number of outliers detected for each one. The ranges for calibration and validation are shown in Table 4.

**Table 4.** Statistical performance of calibration (C) and internal validation (V) models of different analytes.

| Analyte | Pre-Treatment [1] | PC [2] | n [3] (C/V) | Range (C/V) | Q-Value [4] | SEC [5] | SEP [6] | r [7] (C/V) | R² [8] (C/V) | Bias [9] (C/V) | SD [10] (C/V) |
|---|---|---|---|---|---|---|---|---|---|---|---|
| Ash | n01 [11], dg1 [12] | 4 | 180/94 | 40.8–112.8/48.4–107.8 | 0.66 | 0.6 | 0.65 | 0.91/0.90 | 0.83/0.81 | 0/−0.07 | 0.60/0.65 |
|  | mf | 4 | 180/94 | 40.8–112.8/48.4–107.8 | 0.53 | 0.99 | 0.95 | 0.72/0.79 | 0.52/0.62 | 0/−0.23 | 1.00/0.95 |
| CW | mf | 3 | 190/102 | 670.1–916.4/672.0–901.2 | 0.65 | 1.88 | 1.86 | 0.93/0.94 | 0.97/0.89 | 0/0.15 | 1.88/1.86 |
|  | sa3 [13] | 3 | 190/102 | 670.1–916.4/672.0–901.2 | 0.65 | 1.85 | 1.86 | 0.93/0.94 | 0.87/0.89 | 0/0.23 | 1.85/1.86 |
| NDF | sa3 | 4 | 200/94 | 623.6–879.2/631.7–847.2 | 0.64 | 1.72 | 1.51 | 0.95/0.96 | 0.91/0.92 | 0/−0.14 | 1.72/1.51 |
|  | mf | 4 | 200/94 | 623.6–879.2/631.7–847.2 | 0.64 | 1.72 | 1.54 | 0.95/0.96 | 0.91/0.92 | 0/−0.23 | 1.72/1.54 |
| ADF | dg1, n01 | 2 | 190/94 | 379.0–611.5/389.2–588.2 | 0.43 | −0.87 | 3.46 | 0.81/0.79 | 0.65/0.63 | 0/−0.58 | 3.16/3.46 |
|  | mf | 2 | 190/94 | 379.0–611.5/389.2–588.2 | 0.42 | −0.99 | 3.76 | 0.75/0.75 | 0.56/0.56 | 0/−0.87 | 3.58/3.76 |
| ADL | sa3, dg2 [14], SNV [15] | 4 | 190/90 | 38.8–104.8/45.7–102.0 | 0.53 | 0.88 | 0.97 | 0.80/0.78 | 0.65/0.61 | 0/0.09 | 0.88/0.97 |
|  | mf | 4 | 190/90 | 38.8–104.8/45.7–102.0 | 0.45 | 1.23 | 1.26 | 0.56/0.59 | 0.31/0.34 | 0/0.21 | 1.23/1.25 |
| ABL | dg2 | 3 | 200/98 | 61.1–173.9/74.0–164.4 | 0.7 | 0.67 | 0.71 | 0.95/0.92 | 0.90/0.85 | 0/0.03 | 0.67/0.71 |
|  | mf | 3 | 200/98 | 61.1–173.9/74.0–164.4 | 0.58 | 0.9 | 1.02 | 0.90/0.85 | 0.82/0.72 | 0/0.07 | 0.90/1.02 |
| IVDMD | sa3, SNV | 5 | 188/96 | 380.6–791.6/393.5–767.7 | 0.59 | 3.34 | 3.28 | 0.93/0.92 | 0.86/0.86 | 0/0.06 | 3.34/3.28 |
|  | mf | 5 | 188/96 | 380.6–791.6/393.5–767.7 | 0.55 | 3.49 | 3.63 | 0.92/0.91 | 0.85/0.82 | 0/0.12 | 3.49/3.63 |
| IVNDFD | sa3, dg1 | 4 | 188/92 | 298.9–740.2/334.7–724.9 | 0.57 | 3.62 | 3.6 | 0.92/0.91 | 0.84/0.83 | 0/−0.52 | 3.62/3.59 |
|  | mf | 4 | 188/92 | 298.9–740.2/334.7–724.9 | 0.5 | 4.29 | 4.44 | 0.88/0.86 | 0.77/0.74 | 0/−0.38 | 4.29/4.44 |

[1] Transformations; [2] Principal Components–number of factors; [3] Sample number; [4] Qualitative calibration; [5] Standard error of calibration; [6] Standard error of prediction; [7] Correlation; [8] Coefficient of determination; [9] Systematic error; [10] Standard deviation; [11] Normalization between 0 and 1 Multiplicative scatter correction full; [12] First derivative; [13] Smoothing with 3 points; [14] Second derivative; [15] Standard normal variate.

The lowest Q-values detected were 0.43, 0.53, 0.59, 0.57, for ADF, ADL, IVDMD, and IVNDFD, respectively. Higher Q-values were detected for Ash, CW, NDF, and ABL analytes, with values of 0.66, 0.65, 0.64, and 0.7, respectively. None of the calibrations had a Q-value greater than 0.75. The SEC range was from 0.6 to 3.62, for ash and IVNDFD, respectively. Similarly, the SEP range was from 0.65 to 3.6, for ash and IVNDFD, respectively.

Pearson correlation coefficient values between 0.72 and 0.95 were detected for calibration models. For internal validation models, values of 0.78 and 0.96 were detected. The worst values were for ADF and ADL analytes (r < 0.82), while the best values were for Ash, CW, NDF, ABL, IVDMD, and IVNDFD (r > 0.9). The determination coefficients ($R^2$) followed the Pearson correlation coefficients. For ADF and ADL, the values of $R^2$ were below 0.7. For ash, ABL, IVDMD, and IVNDFD the $R^2$ values were above 0.9. The highest $R^2$ values detected were for PC and NDF.

Regarding internal validation bias, the ABL analyte had the smallest bias (Bias = 0.03), followed by IVDMD, Ash, ADL, NDF, and CW. High values were detected for ADF (Bias = −0.58) and IVNDFD (Bias = −0.52). Standard deviations ranged from 0.48 to 3.62.

### 3.3. External Validation of NIRS Calibration Models

3.3.1. Statistical Performance of External Validation Assessed by NIRS

The highest offset value was detected for CW analyte (offset = 12.17), followed by ADF, NDF, IVDMD, ABL, and IVNDFD (Table 5). Below 1.0 values were detected for Ash and ADL analytes. The worst slope values were detected for CW, ADL, and ABL analytes, with values of 0.85, 0.81, and 0.87, respectively.

**Table 5.** Descriptive analyses and statistical performance of external validation of *Urochloa* spp. samples (*n* = 20) performed using near-infrared spectrophotometer (NIRS).

| Propriedade | Mean | Minimum | Maximum | SD [1] | Offset | Slope | RMSEP [2] | SEP [3] | RSD [4] | Bias |
|---|---|---|---|---|---|---|---|---|---|---|
| Ash | 73.3 | 47.2 | 102.5 | 15.73 | −0.59 | 1.09 | 1.02 | 1.03 | 1.04 | 0.08 |
| Cell wall | 796.1 | 701.1 | 865.4 | 47.53 | 12.17 | 0.85 | 2.01 | 2.03 | 1.93 | 0.03 |
| Neutral detergent fiber | 741.3 | 641.9 | 803.3 | 48.66 | −3.28 | 1.00 | 3.77 | 2.03 | 2.06 | −3.19 |
| Acid detergent fiber | 470.8 | 364.7 | 547.1 | 47.33 | −5.29 | 1.07 | 3.16 | 2.46 | 2.48 | −2.02 |
| Acid detergent lignin | 68.8 | 51.9 | 95.8 | 12.25 | 0.81 | 0.81 | 1.06 | 0.93 | 0.91 | −0.52 |
| Acetyl bromide lignin | 121.9 | 69.9 | 155.3 | 21.29 | 2.61 | 0.87 | 1.41 | 0.86 | 0.84 | 1.12 |
| IVDMD | 639.4 | 491.6 | 784.6 | 76.37 | −2.87 | 1.04 | 3.67 | 3.71 | 3.75 | −0.34 |
| IVNDFD | 567.1 | 398.8 | 774.7 | 92.37 | −1.94 | 1.01 | 3.64 | 3.46 | 3.51 | −1.30 |

[1] Standard deviation; [2] Root Mean Square Error of Prediction; [3] Standard error of prediction; [4] Residual standard deviation.

Regarding external validation, the RMSEP value is one which depicts a more complete deviation analysis compared to SEP and RSD. Values of RMSEP above 3 were detected for NDF, ADF, IVDMD, and IVNDFD. Intermediate values (>1 and <3) were detected for ash, CW, ADL, ABL, being the lowest value detected for ash (RMSEP = 1.02). The bias analysis demonstrates that NDF, ADF, ABL, and IVNDFD had the worst results.

3.3.2. Correlation between SRM and NIRS Methods

The correlations between chemical and nutritional values of *Urochloa* spp. assessed using SRM and NIRS are presented in Table 6. Pearson correlation coefficients were higher than +0.75 for all variables, indicating a strong positive correlation between the methods. Nonetheless, the ADL presented the lowest coefficient (r = +0.7518) among all analytes studied.

**Table 6.** Pearson's correlation between standard reference methods and NIRS method in *Urochloa* spp. samples (*n* = 20).

| Item | r [1] | *p*-Value |
|---|---|---|
| Dry matter | 0.8883 | <0.0001 |
| Ash | 0.8463 | <0.0001 |
| Cell wall | 0.9062 | <0.0001 |
| Neutral detergent fiber | 0.9079 | <0.0001 |
| Acid detergent fiber | 0.8925 | <0.0001 |
| Acid detergent lignin | 0.7518 | <0.0001 |
| Acetyl bromide lignin | 0.9227 | <0.0001 |
| IVDMD [2] | 0.8984 | <0.0001 |
| IVNDFD [3] | 0.9314 | <0.0001 |

[1] Pearson's correlation coefficient; [2] In vitro dry matter digestibility; [3] In vitro neutral detergent fiber digestibility.

### 3.3.3. Association between SRM and NIRS Methods

In Figure 2, the regressions between SRM and NIRS analysis for external validation samples are presented. In ascending order, the coefficients of determination ($R^2$) between the SRM and NIRS were: ADL = 0.57; ash = 0.72; ADF = 0.80; IVDMD = 0.81; CW = 0.82; NDF = 0.82; ABL = 0.85; and IVNDFD = 0.87. Excepting ADL, all $R^2$ values were above 0.71.

The angular coefficients (slopes) were approximately, in ascending order: ADL = 0.69; ash = 0.71; IVDMD = 0.76; FDA = 0.80; NDF = 0.86; DIVFDN = 0.92; LBA = 0.95, and CW = 0.97. It is observed that the lowest slope value was detected for LDA, and the others were between 0.71 and 0.97. Analysis to verify an intercept = 0 and slope = 1 was performed for each regression equation. The variables ash, NDF, ADF, ADL, ABL, and IVDMD had intercepts ≠ 0 and slopes ≠ 0 (*p* < 0.05). On the other hand, CW and IVNDFD achieved intercepts = 0 and slopes = 0 with *p* = 0.9642 and 0.068, respectively.

In Table 7 the analysis of variance of each regression model is presented. In this study, more procedure steps are required by SRM as the error was greater than the mean squared (MSE), excepting ash, which had a high MSE was similar to the fiber analysis. Cell wall, NDF, and ADF had MSE values above 400. Regarding lignin methods, ADL had the lowest MSE value followed by ABL. In vitro digestibility had values expressively high compared to chemical variables, showing values of 1187 and 1193, for IVDMD and IVNDFD, respectively.

**Table 7.** Analysis of variance. coefficient of determination ($R^2$). and coefficients of linear regression models between standard reference methods and NIRS of different analytes of *Urochloa* spp. grass (*n* = 20).

| Item | Ash | CW | NDF | ADF | ADL | ABL | IVDMD | IVNDF |
|---|---|---|---|---|---|---|---|---|
| DF$_{residual}$ [1] | 18 | 18 | 18 | 18 | 18 | 18 | 18 | 18 |
| MSE [2] | 74.18 | 426.4 | 438.9 | 480.9 | 68.90 | 75.25 | 1186.8 | 1192.7 |
| $R^2$ [3] | 0.7162 | 0.8213 | 0.8243 | 0.7966 | 0.5651 | 0.8514 | 0.8072 | 0.8675 |
| $R^2_{adjusted}$ | 0.7004 | 0.8113 | 0.8146 | 0.7853 | 0.541 | 0.8432 | 0.7965 | 0.8602 |
| RMSE [4] | 8.61 | 20.6 | 20.9 | 21.9 | 8.30 | 8.67 | 34.4 | 34.5 |
| Coefficients | | | | | | | | |
| Intercept | 20.60 | 22.93 | 128.00 | 110.48 | 25.26 | −5.21 | 158.94 | 632.709 |
| *p*-value | 0.019 | 0.790 | 0.071 | 0.019 | 0.013 | 0.657 | 0.010 | 0.195 |
| Slope [5] | 0.709 | 0.971 | 0.862 | 0.798 | 0.686 | 0.949 | 0.759 | 0.917 |
| *p*-value | <0.0001 | <0.0001 | <0.0001 | <0.0001 | 0.0001 | <0.0001 | <0.0001 | <0.0001 |

[1] Degree of freedom; [2] Mean squared error; [3] Coefficient of determination; [4] root-mean squared error; [5] Angular coefficient.

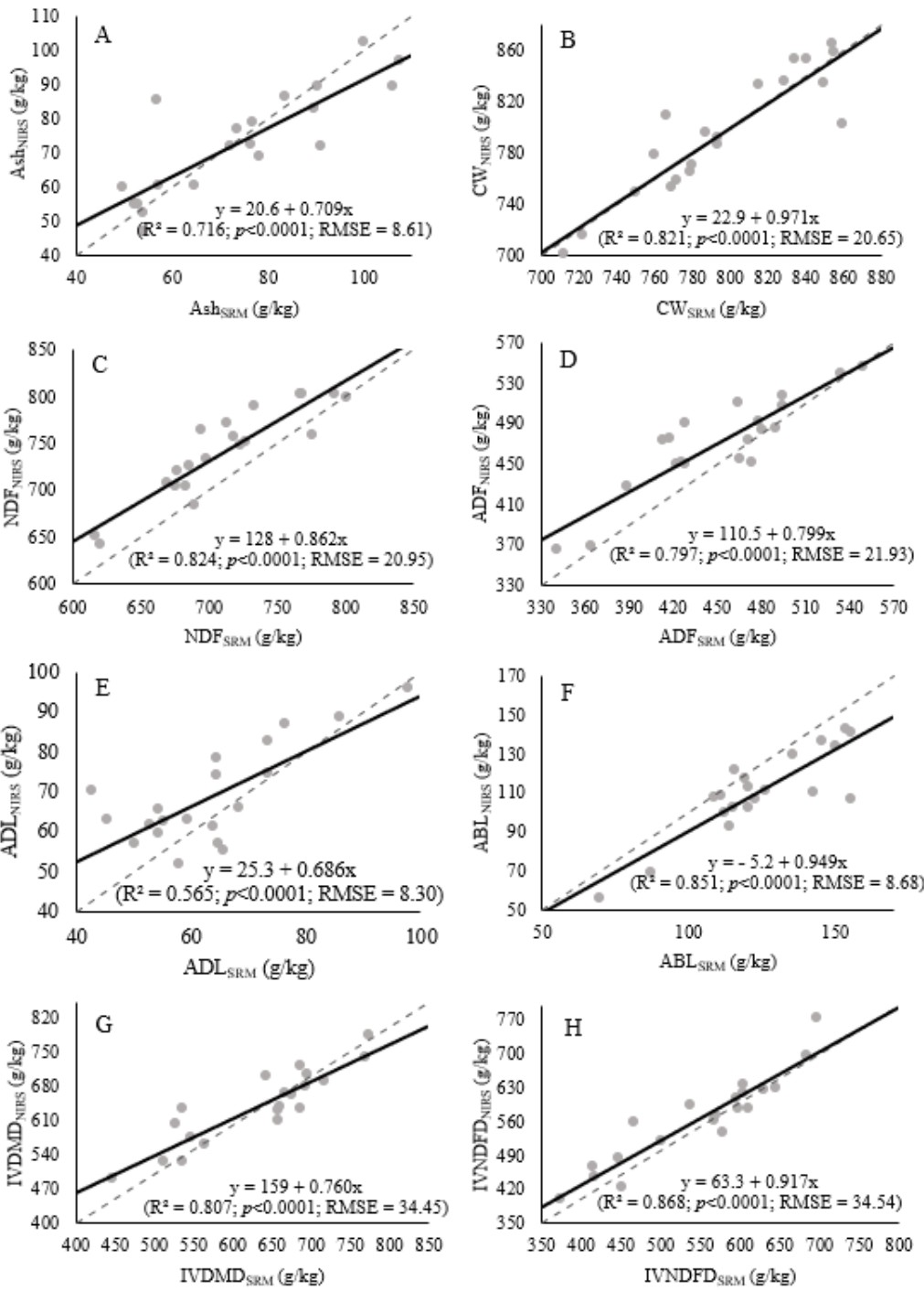

**Figure 2.** Relationship between chemical and nutritional contents of *Urochloa* spp. grass measured using the standard reference methods (SRM; x axis) and values predicted using NIRS (y axis) in the external validation sets; (**A**) ash; (**B**) cell wall; (**C**) NDF; (**D**) ADF; (**E**) ADL; (**F**) ABL; (**G**) in vitro dry matter digestibility; and (**H**) in vitro neutral detergent of fiber digestibility. The black solid line is the relationship between measured and predicted values in the validation sets for quality parameters. The dashed line is the bisector (Y = X). The gray points indicate each observation.

### 3.4. Lignin NIRS-Prediction through ABL or ADL Methods

Pearson correlation between lignin methods and IVDMD and IVNDFD were assessed to determine the best digestibility predictor (Table 8). The ABL method performed by SRM or predicted using NIRS was the best correlated methods (r > 0.80; *p* < 0.01) to assess

in vitro digestibility of both DM and NDF. The ADL method (using SRM or NIRS) had a poor Pearson coefficient compared to ABL methods (r < 0.79; *p* < 0.01).

**Table 8.** Pearson's correlation between lignin content (ADL or ABL) and in vitro dry matter digestibility (IVDMD) and in vitro neutral detergent fiber digestibility (IVNDFD) in *Urochloa* spp. samples performed using wet-chemistry or NIRS (*n* = 20).

| Item | Lignin Methods | | | |
|---|---|---|---|---|
| | **ADL** | **ADL$_{NIRS}$** | **ABL** | **ABL$_{NIRS}$** |
| IVDMD | −0.7691 ** | −0.6685 * | −0.8887 ** | −0.8830 ** |
| IVNDFD | −0.7630 ** | −0.6896 * | −0.8935 ** | −0.8863 ** |
| IVDMD$_{NIRS}$ | −0.7563 * | −0.5816 * | −0.8075 * | −0.9041 ** |
| IVNDFD$_{NIRS}$ | −0.7860 ** | −0.6444 * | −0.9096 ** | −0.9683 ** |

\* *p*-value < 0.01. \*\* *p*-value < 0.0001.

## 4. Discussion

### 4.1. Descriptive Analyses

In an on-farm environment, ruminant nutritionists usually require fast and precise chemical and nutritional forage analysis for their diet formulations. Furthermore, in forage genetic improvement, based on the selection of nutritional traits, minimal chemical details detected using standard reference methods of feed analysis, as well as high-throughput analytical methods, essential for studies involving an elevated number of samples, need to be assessed reliably. Therefore, to reach these requirements, a precise and efficient analytical method, such as NIRS, becomes necessary. Several studies have recently reported the success of NIRS in screening forage quality parameters [16,30–33], combining speed, accuracy, and relatively inexpensive prediction of forage quality parameters [34].

Regarding forage analysis, fiber content, such as CW, NDF, ADF, as well as their composition, which may negatively impact its digestibility, depicted by the lignin content, is the main component in which the nutritionists are interested [35,36]. In this study, the main critical quality parameters, comprising CW, NDF, ADF, lignin, and digestibility, were evaluated for *Urochloa* spp. grass using NIRS. In addition, two methods of lignin quantification were evaluated. The results herein presented due to 149 *Urochloa* spp. accessions at different development stages, showed that the range of most parameters was broader than the range reported by [16,37–39], who studied tropical forages. This represents an outspread implementation of our NIRS models. Among several recommendations stated by ASTM [40], the NIRS must have in its database the largest possible range of results. It is because, after calibration, if samples present results out of this range, the robustness of the result may be impaired. For this reason, over time, that there is also the possibility of adding new samples to the calibration database, to account for wider variability.

### 4.2. NIRS Model Calibrations and Internal Validation

The scope of a mathematical model evaluation is to indicate the accuracy, precision, and robustness of its prediction. In addition, this evaluation involves identifying model weaknesses and indicating those that should not be used [41].

Regarding the range of each analyte, it is noted that in all of them, the calibrations are broader than the internal validations, in agreement with the recommendation, which states that the spectra with minimum and maximum concentration values (extreme values) must be selected as calibration spectra [40]. Nonetheless, according to the low Q-value detected in ADF, ADL, IVNDFD, and IVDMD analysis (Q-value < 0.6), some considerations must be inferred before their use, such as adding samples to the calibration set, modifying pre-treatments, and altering the wavelength prior to selection. It is because the lowest acceptable Q-value is 0.6. More reliable analyses were detected for NDF, CW, Ash, and ABL (Q-value > 0.6). However, no calibration model had a Q-value above 0.75, which is the lowest Q-value required for a reliable model. Accurate NIRS predictions depend on three

conditions: (1) a standard reference method with low laboratory error; (2) chemical bonds in the sample capable of being absorbed or identified by NIRS; and (3) significant quantity and large amplitude of the constituent to be analyzed [42]. To summarize, all analytes with low Q-value have methodological limitations. For example, ADF and ADL content may be underestimated by the loss of potentially soluble lignin in the ADF method [15]. According to [14], about 50% of lignin was lost by using the ADL method in tropical grasses. Accordingly, IVNDFD and IVDMD require multiple steps, which may interfere with the accuracy of the result, transferring the SRM error to NIRS.

For calibration inspection, SEC and SEP parameters provide the magnitude of the standard deviation for calibration and validation (precision), respectively [40]. Both values should be as small and as similar as possible. They could be compared to the standard deviation of SRM [40]. The ADF had the worst consistency of deviation value, requiring enhancement of its model before use. The reason for this poor result may be the same abovementioned; lignin loss by the acid detergent method. Otherwise, [43] studying temperate grasses had better SEC values compared to this study: 0.43, 0.21, and 1.14, for NDF, ADL, and IVDMD, respectively. Likewise, [44], studying alfalfa (*Medicago sativa*, L.), reached SEC values of 1.06, 0.79, and 0.31, for NDF, ADF, and ADL, respectively. Correlation coefficients detect how well the predicted (NIRS) values fit the reference (laboratory) values, on average. They must be at least greater than 0.9 [40]. Determination coefficients followed the correlation coefficients with ADF and ADL having the lowest coefficient values compared to all other analytes.

The study performed by Simeone et al. [20] also used several genera of *Urochloa* spp., under similar climate conditions as this study. The authors detected better coefficients of determination, with $R^2$ equal to 0.95 for NDF, 0.93 for ADF, 0.94 for ADL, and 0.94 for IVDMD. In another study with *Brachiaria humidicola*, [45] detected r values for calibration equal to 0.89 for NDF and ADL, 0.81 for ADF, and 0.96 for IVDMD, close to the present results. In the same work, with cross-validation, they obtained $R^2$ equal to 0.89 for NDF, 0.81 for ADF, 0.89 for ADL, and 0.86 for IVDMD. The results were better for ADF and ADL, similar for IVDMD, and lower for NDF, when compared to the present experiment.

In NIRS, accuracy is measured by the bias of the internal validation set (V-Set Bias). It provides information on the average deviation between predicted (NIRS) and actual (laboratory) values, which should always be as close to 0 as possible. This value provides information about the systematic deviation of calibration [40]. The bias of the calibration set (C-Set Bias) is zero by definition [40]. High bias values were detected for ADF and IVNDFD. As previously mentioned, the methodological approach as well as the number of laboratory steps in the analytical method may negatively influence the data dispersion, jeopardizing the calibration model.

### 4.3. External Validation of NIRS Calibration Models

High RMSEP values were detected for NDF, ADF, IVDMD, and IVNDFD. However, only the ADL analyte presented a lower Pearson's correlation coefficient. This result for ADL may be due to the laboratory coefficient of variation, which among all analyses, was the highest. Regardless of whether by sub-sampling or using the procedure with low analytical precision, it is known that errors arising from SRM are transported to NIRS. Freitas et al. [45] found a coefficient of correlation of 0.89, 0.81, 0.89, and 0.82, for NDF, ADF, ADL, and IVDMD, respectively, with 48 samples of *Brachiaria humidicola*. Clark and Lamb [46] reported results between 0.76 and 0.84 for IVDMD, in several types of grass. Other researchers also showed correlations similar to those found in this study [16,47]; however, it is difficult to compare studies in this sense, because comparisons should be performed using the same material. The coefficient of correlation is reflected in the coefficient of determination, in which ADL had the lowest value. Through linear simple regressions, it was detected that even ADL is a poor method for calibration and internal validation in NIRS; the RMSE showed that NIRS was able to adjust its calibration values presenting acceptable values in external validation.

*4.4. Lignin NIRS-Prediction through ABL or ADL Methods*

The measurement of lignin is a powerful tool from a nutritional perspective. Assessing lignin content, the forage nutritional value can be assessed by estimating its digestibility [48]. The digestibility of both DM and NDF are negatively influenced by the lignin content present in the feed [11]. Thus, ruminal digestion evaluation is more important than lignin content evaluation [49].

It is known that there is no method of choice to measure lignin in forage and it is also well accepted that the higher the lignin content in a plant sample, the lower its digestibility [50,51]. Therefore, Pearson correlations between the methods used to quantify lignin and digestibility were assessed to try to clarify this issue. As expected, all results were negatively correlated, for both SRM and NIRS methods. Corroborating Fukushima and Hatfield [52], the results herein related to ADL and ADL$_{NIRS}$ generated lower Pearson correlation coefficients r = $-0.58$ to $-0.78$ than those assessed using ABL, with Pearson correlation coefficients between $-0.80$ and $-0.96$. As mentioned earlier, this fact is likely to be a result of the SRM, before NIRS analyses, or when the data were transferred to the instrument, and not some failure when performing the procedures in the device [53].

## 5. Conclusions

Near-infrared spectroscopy presented potential for the quantification of *Urochloa* spp. properties, such as lignin contents (ADL and ABL) and ash, CW, NDF, ADF, IVDMD, and IVNDFD. However, calibrations performed by NIRS to measure ADF, ADL, IVDMD, and IVNDFD need to be thought about with caution before their utilization as routine analysis. Correlation analysis between in vitro digestibility and lignin content showed that ABL and ABL$_{NIRS}$ methods are better than ADL and ADL$_{NIRS}$ for lignin quantification in *Urochloa* spp. grass.

**Author Contributions:** I.C.d.S.B.G.: Conceptualization, investigation, methodology, writing–original draft preparation T.H.d.S.; writing–original draft preparation, writing–review and editing I.C.d.S.B.G.; writing–review and editing R.S.F.; investigation, methodology I.C.d.S.B.G., T.H.d.S. and C.C.P.; formal analysis R.S.F.; funding acquisition, visualization, supervision. All authors have read and agreed to the published version of the manuscript.

**Funding:** This research was funded by Coordination for the Improvement of Higher Education Personnel (CAPES; Financial code: 001).

**Institutional Review Board Statement:** All the experimental procedures were approved by the Bioethics Committee of the School of Veterinary Medicine and Animal Science, University of São Paulo (approval number: 7684280518).

**Data Availability Statement:** We confirm that we have full access to all the data in the study and take responsibility for the integrity of the data and accuracy of the data analysis. The data supporting the findings of this study are available from the corresponding author upon reasonable request.

**Acknowledgments:** The authors express appreciation to the School of Veterinary Medicine and Animal Science from University of Sao Paulo for providing staff and facilities. In addition, the authors thank the National Council for Scientific and Technological Development (CNPq) and the Coordination for the Improvement of Higher Education Personnel (CAPES) for providing fellowships.

**Conflicts of Interest:** The authors declare no conflict of interest.

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
