# Peer review of "Assessing the Repeatability and Reliability of NIRS to Predict Nutritional Values and to Evaluate Two Lignin Methods in Urochloa spp. Grasses"

_2813-3463, doi:10.3390/grasses2020010_

Round 1

Reviewer 1 Report

I suggested some details need to consider 

1.       Urochloa should be italic

2.       In line 17, it should be “evaluating” instead.

3.       Abbreviations, as CW, NDF, ADF and DM, should have their own full forms at their advent.

4.       The result in the abstract was not amphasized enoguh and did not intepret the whole of the study.

5.       Why was Brachiaria spp. not mentioned in the abstract while it appeared in the keyword and the subject of the study was Urochloa spp.instead?

6.       Please specify what regions and climate condtions in Brazil where the samples were collected.

7.       The samples were too mixed, since they belonged to different cultivars and were grathered unevenly from 4 different regions for 169 grass samples, leading to a consequence that the quality of the samples was questionable and they cannot represent for Urochloa spp. grass in the whole Brazil.

8.       Why was the in vitro digestibility carried out on Brachiaria decumbens?

9.       In line 95, it should be “[25]” instead.

10.   The amount of samples used in each experiment and replications were missing.

11.   Why the samples were divided into 149 and 20 samples, in another word, on what criteria, and how were they divided?

12.   Some informations in the discussion need citations.

Author Response

Dear reviewer,

We would like to thank you for your time. The constructive criticisms and thoughtful comments have improved the quality of this manuscript and we sincerely appreciate them. We carefully addressed all comments and revised the manuscript accordingly. 

Please find below the answers (in red) to your comments.

Thank you

Reviewer 2 Report

Dear authors,

Congratulation for your article. Just fewer adjustments are required.

-          Abbreviations when first appear must be preceded by full name.

-          The abstract must show some results and conclusions of the research.

Author Response

(The authors gave the same response as above.)

Reviewer 3 Report

L19: Urochloa may have to spelled in italics.

L22: The article in front of “in vitro digestibility” may be omitted.

L32: In this particular case, carbon sequestration capacity and “great efficiency of nitrogen utilization” are imprecise terms.

L34: What do you mean by “high productivity?”

L40: You may omit the hyphen.

L51: The near-infrared technique…

L29-65: I suggest carefully reviewing and editing the entire introduction

L55: It is not quite true that NIRS doesn’t require sample processing. In fact, if you use a single set then you will have to establish a prediction model using wet chemistry.

L85. The degree symbol needs to be positioned higher.

L85: What are those 4 regions? Those you describe in the previous paragraph? How many samples did you harvest?

Figure 1: Please add a unit to the Y axis and make clear the x axis means wavelength.

L138: Switch the nanometers around so it reflects the order of cm-1.

L160: You don’t have to reintroduce the abbreviations.

Author Response

Dear reviewer,

We would like to thank you for your time. The constructive criticisms and thoughtful comments have improved the quality of this manuscript and we sincerely appreciate them. We carefully addressed all comments and revised the manuscript accordingly. 
